# Perspective of the Relationship between the Susceptibility to Initial SARS-CoV-2 Infectivity and Optimal Nasal Conditioning of Inhaled Air

**DOI:** 10.3390/ijms22157919

**Published:** 2021-07-24

**Authors:** Ranjan Ramasamy

**Affiliations:** ID-FISH Technology Inc., 556 Gibraltar Drive, Milpitas, CA 95035, USA; rramasamy@idfishtechnology.com

**Keywords:** age, COVID-19, gender, genetic factors in SARS-CoV-2 susceptibility, innate immunity, nasal conditioning of inspired air, SARS-CoV-2 infectivity, upper respiratory tract immunity

## Abstract

Severe acute respiratory syndrome coronavirus 2 (SARS-CoV-2), as with the influenza virus, has been shown to spread more rapidly during winter. Severe coronavirus disease 2019 (COVID-19), which can follow SARS-CoV-2 infection, disproportionately affects older persons and males as well as people living in temperate zone countries with a tropical ancestry. Recent evidence on the importance of adequately warming and humidifying (conditioning) inhaled air in the nasal cavity for reducing SARS-CoV-2 infectivity in the upper respiratory tract (URT) is discussed, with particular reference to: (i) the relevance of air-borne SARS-CoV-2 transmission, (ii) the nasal epithelium as the initial site of SARS-CoV-2 infection, (iii) the roles of type 1 and 3 interferons for preventing viral infection of URT epithelial cells, (iv) weaker innate immune responses to respiratory viral infections in URT epithelial cells at suboptimal temperature and humidity, and (v) early innate immune responses in the URT for limiting and eliminating SARS-CoV-2 infections. The available data are consistent with optimal nasal air conditioning reducing SARS-CoV-2 infectivity of the URT and, as a consequence, severe COVID-19. Further studies on SARS-CoV-2 infection rates and viral loads in the nasal cavity and nasopharynx in relation to inhaled air temperature, humidity, age, gender, and genetic background are needed in this context. Face masks used for reducing air-borne virus transmission can also promote better nasal air conditioning in cold weather. Masks can, thereby, minimise SARS-CoV-2 infectivity and are particularly relevant for protecting more vulnerable persons from severe COVID-19.

## 1. Background to SARS-CoV-2 and COVID-19

Coronavirus disease 2019 (COVID-19) due to the severe acute respiratory syndrome coronavirus 2 (SARS-CoV-2) is a pandemic that, since its first identification in December 2019, caused approximately 178.5 million infections and 3.9 million deaths worldwide by 21 June 2021 [1]. SARS-CoV-2 is a membrane-enveloped virus with a 30 kb positive-sense RNA genome. It is related to two highly pathogenic coronaviruses of zoonotic origin that previously triggered limited disease outbreaks, SARS-CoV-1 in 2002–2004 and the Middle East respiratory syndrome coronavirus sporadically since 2012 [2]. 

SARS-CoV-2 is also related to several less pathogenic coronaviruses that cause mild to moderate common cold-like symptoms in up to 30% of people every year [2]. Early SARS-CoV-2 infection is commonly diagnosed by RT-qPCR for viral RNA in the nasal cavity and nasopharynx. The spike glycoprotein (S) located on the SARS-CoV-2 membrane envelope mediates binding of the virus to epithelial cells of the respiratory tract to initiate infection. 

S is composed of an N-terminal S1 region containing a receptor binding domain that attaches to the angiotensin-converting enzyme 2 (ACE2) receptor on host cells and a C-terminal S2 region that subsequently mediates fusion between the virus and host cell membranes to allow the entry of viral RNA into cytoplasm [2,3]. SARS-CoV-2 can also enter cells by endocytosis followed by S-mediated fusion of the endosome and virus membrane [4]. S also facilitates cell-cell membrane fusion that additionally spreads virus [4].

Transfer to the mucous membranes of the eyes, nose, and mouth by fomite or direct contact is an established transmission method for respiratory viruses. It has increasingly become clear that nasal inhalation of virions present in exhaled breath and airborne droplets produced by sneezes and coughs of infected persons is a major route of infection for SARS-CoV-2 [5,6,7,8,9]. Once infected with SARS-CoV-2, most people, typically healthy young individuals, develop mild or no symptoms because they rapidly eliminate the virus from the upper respiratory tract (URT) through an effective immune response [10,11]. 

However, SARS-CoV-2 infection causes severe pneumonia in about 15% of patients and acute respiratory distress syndrome (ARDS), which is difficult to treat, in about 5% of patients [11]. The earliest immune response to the infection of airways by a respiratory virus, such as influenza A, is mainly innate and antigen-nonspecific; however, this is rapidly followed by an adaptive immune response involving antigen-specific B and T lymphocytes [12,13,14]. Accumulating evidence suggests that the immune response to SARS-CoV-2 infection follows a similar course [10,15]. Overactive and inappropriate adaptive and innate immune responses that ensue if SARS-CoV-2 is not eliminated early in the URT contribute to the characteristic immunopathology of ARDS and severe COVID-19 with lung and systemic involvement [10,11].

The incidence of many respiratory viral infections, including those caused by influenza and respiratory syncytial viruses, increases during winter in temperate zone countries. Preventive measures, such as vaccination, are therefore always undertaken before the onset of winter to mitigate influenza epidemics. The winter peak of infections has generally been attributed to the better environmental survival of the influenza and respiratory syncytial viruses at cold temperatures and increased opportunities for transmission when people spend more time indoors in winter [16,17,18]. 

An upsurge in winter infections is also characteristic of seasonal coronaviruses and generally the case with SARS-CoV-2 [19], although factors, such as population immunity and the emergence of more transmissible variants of SARS-CoV-2, can modify the relationship between the incidence of disease and environmental conditions. Other respiratory viruses have different transmission seasons with some transmitted throughout the year [18]. Striking outbreaks of COVID-19 among people working for prolonged periods at low ambient temperatures in meat and poultry processing factories occurred year round in many countries [20]. This has been attributed to crowded working conditions, but it was recognised more recently that low working temperatures may increase the risk of COVID-19 [20]. Influenza tends to be more prevalent during the rainy season in tropical countries, and this has been ascribed to greater congregation indoors during the rains [16,17]. It is also pertinent, however, that ambient temperatures are somewhat lower and the humidity is higher during the rainy season in the tropics, and these changes can be more pronounced at higher elevations. Mathematical analyses showed that the rates of exponential spread of SARS-CoV-2, rather than COVID-19 morbidity and death, correlated best with environmental temperature in northern temperate zone countries [21,22].

People with recent tropical ancestry are more prone to severe COVID-19 than persons of temperate zone ancestry in the UK and USA, with the difference attributed largely to socio-economic factors [23,24]. COVID-19 also results in more severe disease in elderly persons, and thus the relative risk of dying from COVID-19 increases exponentially with age in many countries [25]. This has been ascribed to age-related changes in the immune system reducing the ability to mount an effective immune response against SARS-CoV-2 and the increasing prevalence of other morbidities with age facilitating more severe COVID-19 [26,27]. Fatality rates following infection are also generally higher in males compared with females in all age groups [25,27].

An early unpublished observation that nasal warming and humidifying of inspired air (nasal air conditioning) may influence protective immune responses in the URT to SARS-CoV-2 infection [28] is now evaluated in detail with recently published data.

## 2. SARS-CoV-2 Infection in the Upper Respiratory Tract 

An early virological study during the COVID-19 pandemic suggested that SARS-CoV-2 first infected and replicated in the nasopharynx and oropharynx of the URT, with the likely subsequent seeding of the lower respiratory tract and lungs by aspiration, although infection of the nasal epithelium was not investigated in this work [8]. Subsequent molecular studies demonstrated that ACE2 expression and SARS-CoV-2 infection are higher in the nasal epithelium than the lower respiratory tract, and therefore that the nasal epithelium is the probable initial infection site followed by infection of the pharynx as a result of mucociliary clearance of virus towards the nasopharynx and later the likely seeding of the lower respiratory tract and lungs by aspiration [29,30,31]. This is also consistent with the presence of TMPRSS2, the protease that cleaves S to expose its fusion peptide, in the nasal epithelium [30]. The viral load of SARS-CoV-2, indicative of viral replication, is greater in the nasal epithelium compared with in the pharynx following infection [31]. The nasal epithelium is an established site for the replication and transmission of influenza viruses [32]. Influenza viruses replicating in the nasal epithelium have been shown to reach the pharynx through mucocilary clearance and serve as a source of virus for subsequently infecting lungs [32] through the oral-lung axis [33]. Although definitive experimental evidence is not yet available, it is reasonable to assume that the same process also occurs with SARS-CoV-2.

Current data suggest that infection with SARS-CoV-2 in most people typically produces either mild or no symptoms of COVID-19 because the innate and adaptive immune responses are able to rapidly eliminate the virus from the URT [10], and that severe disease with lung involvement requiring hospitalisation occurs only when such immune responses are delayed or inadequate [10,11]. The susceptibility to infection with the virus is difficult to distinguish from susceptibility to severe COVID-19 because of varying disease phenotypes and multiple factors influencing the propensity to develop severe disease [10,11]. Early RT-qPCR test positivity in the URT is possibly the best presently available criterion for detecting early infection and is, therefore, useful in estimating susceptibility to infection and differentiating it from susceptibility to severe COVID-19. This article only considers the role of nasal air conditioning on susceptibility to infection with SARS-CoV-2 in the URT and not the many other factors that subsequently determine the development of severe COVID-19, which may follow the URT infection.

## 3. Physiological Importance of the Nasal Conditioning of Inspired Air

Healthy adults exchange approximately 1–1.5 × 10^4^ litres of air per day with the environment through inspiration and expiration [34,35]. The nasal air conditioning of inspired air before it reaches the lungs is essential for healthy respiratory function [34,35]. The nasal mucosa possesses a specialised sub-epithelial network of capillaries to support air conditioning so that inspired air of approximately 25 °C and 35% relative humidity is warmed and humidified to about 33–34 °C and about 90% relative humidity before it enters the nasopharynx [34,35]. 

Further warming to the alveolar temperature of 37 °C and relative humidity of 100%, which is critically important for lung function [34,35], normally takes place in the rest of the respiratory airway through heat and moisture exchange with its mucosal surface. Conversely, air from the lungs loses humidity and warmth while progressing up the respiratory tract for expiration [34,35]. Nasal air conditioning capability varies with the temperature and humidity of inspired air [34,35]. Even small increases in the temperature of the nasal mucosa enhances the ability of the nose to adequately warm and humidify inspired air [36].

## 4. Innate and Adaptive Immune Response in the Upper Respiratory Tract in Protection against SARS-CoV-2 Infection

Protective antigen non-specific or innate immune mechanisms operative in the respiratory tract against viral infections have been recently reviewed [37]. Such innate immune mechanisms have been best characterised in influenza [12,13,14]. Analogous innate immune mechanisms that can protect against infection by SARS-CoV-2 in the URT are summarised in Table 1.

The adaptive immune response also plays a role in clearing SARS-CoV-2 infections in the airways and is responsible for the enhanced protection in the URT conferred through vaccination or a prior resolved infection with SARS-CoV-2. The adaptive immune response that controls and eliminate infection in the URT is induced by viral antigens reaching the nasopharynx-associated lymphoid tissue (NALT) for presentation to T and B lymphocytes. The efficacy of this process is illustrated by the elicitation of protective secretory IgA in the nasal mucosa as well as protective systemic T cell and antibody responses by intranasally administered influenza vaccines [38]. Blood IgG antibody levels and CD8+ cytotoxic lymphocyte responses were correlated with protection against a second infection with SARS-CoV-2 in macaques [39]. Multifunctional antibodies with virus neutralising, complement activating, phagocytosis promoting, and NK-cell-activating properties as well as interferon γ (IFNγ, a Type 2 IFN) producing T cells are robustly induced after vaccination with S, and such immune responses are associated with vaccine efficacy [40,41,42,43,44,45]. Serum IgA antibodies to S are induced by the vaccines; however, it is not yet known how this relates to the production of dimeric secreted IgA in the URT mucosa, which is important for conferring protection against influenza virus infection [12,13,14]. Antibody and T cell responses in COVID-19 and their functions in resolving disease are increasingly becoming clarified [10,15,46,47,48]. The principal adaptive immune mechanisms that can protect against infection by SARS-CoV-2 in the URT are summarised in Table 2.

## 5. Humidity of Inspired Air and Protection against SARS-CoV-2 Infection

Mucus produced by goblet cells forms a layer covering the ciliated epithelium of the URT and is the first barrier that needs to be overcome by respiratory viruses in order to infect epithelial cells. The mucosal barrier functions optimally at 100% relative humidity and the core body temperature of 37 °C [35]. In mice infected with influenza A virus, low air humidity impairs the mucociliary clearance of virions, Type 1 IFN-dependent antiviral defence in epithelial cells, and the repair of damaged epithelium [49]. 

The multiple ways in which humidity may affect the stability of respiratory viruses and the mucosal barrier function have been recently reviewed [37] and may be summarized as follows for the URT: (i) airborne enveloped viruses may be more stable at low and high humidities and less stable at intermediate humidities, (ii) mucoepithelial integrity is decreased by inspired air of low humidity, and (iii) mucocilary clearance, which removes virions trapped in the mucus from the airway, is reduced at low humidity. Inadequate humidification of inhaled air of relatively low moisture content that is characteristic of winter may, therefore, be expected to enhance infection of the URT epithelium by SARS-CoV-2.

## 6. Temperature of Inspired Air and Protection against SARS-CoV-2 Infection

Low temperature affects the stability of respiratory viruses in the environment and also compromises mucosal barrier function [37,49]. There is evidence to suggest that SARS-CoV-2 survives better at lower temperatures in human nasal mucus and sputum [50], which is pertinent to both airborne and fomite transmission. Essentially, it can be surmised that lower than optimal temperatures in the URT: (i) may improve the stability of the lipid bilayer in enveloped viruses, (ii) reduce the mucociliary clearance of viruses, and (iii) compromise URT-epithelium repair during infections [37,49].

Increasing evidence has demonstrated that lower-than-optimal airway temperatures also compromise the critical innate immunity against an initial virus infection that is normally conferred by the production of IFNs in airway epithelial cells [51]. Viral infection of epithelial cells leads to the production of type 1 IFNα and IFNβ as well as type 3 IFNλ. Viral RNA in the cytoplasm, including SARS-CoV-2 RNA, is recognised as a pathogen-associated molecular pattern (PAMP) by at least two prominent PAMP-recognising receptors (PRRs) that are products of the retinoic acid-inducible gene 1, termed RIG-1, and the melanoma differentiation-associated gene 5, termed MDA5. The activation of RIG-1 and MDA5 initiates a signalling cascade that leads to the phosphorylation of two important regulators of type 1 IFN production—termed IFN regulatory factors 3 and 7 (IRF3 and IRF7) [51]. 

SARS-CoV-2 RNA released within endosomes during the alternate endocytic entry pathway activates Toll-like receptors, which are PRRs on endosomal membranes, and these can also lead to the phosphorylation of IRF3 and IRF7 in the cytoplasm as well as the activation of the transcription factor NF-κB, which promotes inflammation. Phosphorylated IRF3 and IRF7 dimerise and translocate to the nucleus to initiate transcription and the secretion of IFNα and IFNβ, which bind to type 1 IFN membrane receptors for both interferons—termed IFNARs—on adjacent cells. IFNARs are composed of two subunits termed IFNAR1 and IFNAR2. Activation of IFNAR through the binding of type 1 IFNs initiates a signalling cascade, which results in the transcription of numerous interferon-stimulated genes (ISGs) that confer a virus infection-resistant state to epithelial cells surrounding the virus infected cell [51]. A double-stranded RNA-dependent protein kinase R (an ISG) (i) phosphorylates an initiation factor eIF-2 to block the translation of viral RNA and (ii) triggers apoptosis of the infected cell. 

Additionally, double-stranded RNA in the cytoplasm stimulates the enzyme 2′5′-oligoadenylate synthase (OAS, an ISG) to produce 2′5′oligoadenylate, which activates a latent endonuclease RNAse L to degrade viral RNA. Type 3 λ IFNs are induced in a similar manner to type 1α and β IFNs but, in comparison with type 1 IFNs, (i) have different IFN receptors that are expressed prominently in barrier epithelial cell membranes, (ii) are less inflammatory, and (iii) show more sustained production [51]. Type 3 λ IFN may, therefore, be particularly important in the earliest stage of SARS-CoV-2 infection in the nasal and pharyngeal epithelium and in clearing the infection with mild or no disease. Several proteins coded for by non-structural genes and open reading frames in the SARS-CoV-2 genome interfere with the induction of type 1 and 3 IFNs and their subsequent signalling pathways [52], which is consistent with the importance of these IFN pathways in resisting viral infection.

A common cold-causing rhinovirus replicated better at 33 °C than at 37 °C in primary mouse airway epithelial cell cultures. Increased replication was associated with the better PRR-mediated induction of type 1 and 3 IFNs as well as ISGs in the epithelial cells by the virus at 37 °C compared with at 33 °C [53]. Experiments on the infection of primary human bronchial epithelial cell cultures with rhinoviruses suggested that the inhibition of both apoptotic cell death and RNAse L may also be responsible for the better rhinovirus replication at 33 °C compared to 37 °C in these cells [54].

Emerging evidence now suggests that SARS-CoV-2 also replicates approximately 100-fold better at 33 °C than at 37 °C in human airway epithelial cells and that this is associated with better induction of type 1 and 3 IFN-mediated ISGs at 37 °C [55]. SARS-CoV-2 growing in human airway epithelial cells in culture were sensitive to inhibition by type 1 and 3 IFNs [56]. As human nasal temperatures are typically maintained at 33–34 °C under normal environmental conditions, while the temperature in the lower respiratory tract is 37 °C [34,35,36], these findings are consistent with observations that SARS-CoV-2 initially infects the nasal epithelium followed by the nasopharynx and only, subsequently, the lower respiratory tract [29,30,31].

Protection conferred by the type 1 IFN pathway is also supported by the recent demonstration that persons with genetic defects in Toll-like receptor 3, a PRR that senses double stranded RNA, as well as genetic defects in IRF7 and the IFNAR1 subunit, were more susceptible to severe, life-threatening COVID-19 pneumonia [57]. Genome-wide association studies (GWAS) in severe COVID-19 also identified genes for the IFNAR2 subunit and OAS as required to protect against critical illness [58]. Severe and life threatening COVID-19 manifests in the lungs and systemically, and hence the above findings may be the result of defective innate immune responses in the lower airways. 

However, such genetic defects will also manifest as suboptimal innate immune responses in the URT. Existing evidence suggests that if SARS-CoV-2 infection in the nasal epithelium and nasopharynx is rapidly eliminated through robust early immune responses, involving type 1 and 3 IFNs, then serious disease in the lungs through subsequent virus seeding will not occur or is minimised [10]. The evidence also suggests that type 1 and 3 IFN production is weaker at lower temperatures in the nasal cavity. 

Colder-than-normal inspired air in winter may be expected to produce temperatures in parts of the nasal cavity that are lower than 33 °C, thereby, likely further facilitating the infectivity of SARS-CoV-2 in the nasal epithelium and nasopharynx. The relationships between the temperature of inspired air, intranasal temperatures, and nasal air conditioning have not been systematically studied to date. However, modelling studies showed that the anterior nasal cavity is responsible for most of the warming of inspired air [59].

The effects inadequate nasal warming of colder inspired air on adaptive and other types of innate immune cell responses remain to be fully ascertained; however, it is expected that they will also be compromised if early innate immune responses and the integrity of the airway epithelium are adversely affected by weaker nasal air conditioning. This can further promote SARS-CoV-2 infection of the nasal epithelium and nasopharynx.

## 7. Nasal Air Conditioning and Genetic Differences in Susceptibility to SARS-CoV-2 Infection

The propensity to develop severe COVID-19 as a result of defects in the innate immune genes for IRF7 and IFNAR1 [57] as well IFNAR2 and OAS [58], which are also likely to affect early innate immune responses in the URT, has been outlined above in Section 6. Additionally, recent GWAS have identified large segments of human chromosomes inherited from Neanderthals that are associated with a major risk for susceptibility [60] or protection [61] against severe COVID-19. These two studies did not identify specific genes responsible for susceptibility or protection but demonstrated a variable distribution of the relevant chromosomal regions in different parts of the world. 

The differential susceptibility and resistance to a variety of human infectious diseases are governed by genetic factors [62] and are particularly well studied in malaria [62,63,64,65]. However, genetic factors that specifically affect the initial infectivity of SARS-CoV-2, in contrast to the severity of ensuing COVID-19, have not been clearly established. The nearest experimental approach to investigate this difference recently examined SARS-CoV-2 RNA test positivity separately from disease phenotype in a US-based GWAS [66]. The results showed that blood group O was significantly associated with reduced SARS-CoV-2 test positivity and an association between some types of tropical ancestry and SARS-CoV-2 test positivity [66]. It is possible to hypothesise that the blood group O association is due to protection conferred in the URT by natural anti-A and anti-B antibodies universally present in blood group O individuals acting against infecting virions carrying membrane A and B antigens derived from infecting individuals of blood groups A and B. 

Variations in nasal structure between human populations living in geographically disperse locations with different climates have been correlated with the greater need to humidify and warm inspired air during cold and dry winters on one hand and the correspondingly reduced need for this in warm and humid climates of the tropics on the other [67,68,69]. It is reasonable to postulate that selection by respiratory viral infections in temperate zones in ancient times may have been a factor that contributed to such nasal variations. SARS-CoV-2 may, therefore, be more infectious in temperate zone countries to persons with a tropical ancestry [66] due to a weaker nasal air conditioning ability, and that this may contribute, in addition to socio-economic factors, to their observed propensity to develop more severe COVID-19 [23,24]. This remains to be definitively established but their potentially greater vulnerability to SARS-CoV-2 infection may be an important consideration for prioritising additional protective measures and vaccination.

## 8. Differences in Nasal Air Conditioning and the Age and Gender Differences in Susceptibility to SARS-CoV-2 Infection

Intranasal air temperature and humidity are lower in elderly in comparison with younger persons, and this is associated with the atrophy of the nasal mucosa with increasing age [70]. Recent radiological studies confirmed such age-related changes [71]. Many nasal parameters also display a pronounced gender dimorphism in diverse populations [68]. Through possible differential nasal air conditioning, these variations may contribute in some small measure to age- and gender-related differences in the susceptibility to infection with SARS-CoV-2, which can then, in turn, impact the frequency of severe COVID-19 [25,26,27]. 

An increase in the basal level of immune activation or inflammation, reduced innate responses in the airway epithelium, deterioration of the quality of adaptive immune responses involving B and T lymphocytes with age, are additional immunological factors [26] that, together with a possibly reduced nasal air conditioning ability, may permit better replication of SARS-CoV-2 in the nasal epithelium and nasopharynx of older persons and, thereby, facilitate severe lower respiratory tract disease and increased mortality [25,26,27]. 

There are also gender-related differences in the innate and adaptive immunity [72] that can play a role in limiting SARS-CoV-2 infection of the URT epithelium, with an attendant impact on the severity of any ensuing COVID-19 [25]. The relative contributions of these other factors and possibly altered nasal air conditioning ability toward the age and gender-related differences in the susceptibility to COVID-19 merit further investigation. It is encouraging, however, that age or gender does not affect vaccine efficacy [40,41,42,43,44,45] do not affect the vaccine efficacy, which also augurs well for immunity generated after recovery from a primary SARS-CoV-2 infection.

## 9. Other Factors Influencing Susceptibility to SARS-CoV-2 Infection

Air pollution can potentially play a role in the susceptibility to SARS-CoV-2 infection by providing air-borne particles to transport virions and affecting the barrier and innate immunity functions of the respiratory epithelium [73]. Available evidence also suggests that the relatively common URT conditions of allergic rhinitis and chronic rhinosinusitis do not increase the risk of COVID-19 [74]. The elevated production of T_H_2 cytokines, which is common in airway allergic diseases, reduces ACE2 but increases TMPRSS2 expression levels in the URT [75]. Rhinitis can, however, promote the transmission of SARS-CoV-2 from infected to uninfected persons as a result of increased nasal mucus production and sneezing. Another likely confounding factor is that prior infection with common cold coronaviruses may generate a degree of cross-reactive protective immunity against SARS-CoV-2 infection in the URT. Cross-reactions between common cold coronaviruses and SARS-CoV-2 have been documented at the level of CD4+ T_H_ cells and CD8+ cytotoxic lymphocytes [46,47]. Vaccination against COVID-19 and previous resolved infections with SARS-CoV-2 augment adaptive immune responses in the URT and will, therefore, reduce the susceptibility to infection. In contrast, the emergence of SARS-CoV-2 variants with a higher affinity of the receptor binding domain of S for ACE2 and reduced binding to neutralizing antibodies [76], as well as potential other mechanisms to evade protective immunity in the URT, will increase the susceptibility to infection. The contribution of the nasal air conditioning ability on SARS-CoV-2 infectivity in the URT also has to be considered in the context of such additional factors.

Figure 1 summarizes different aspects of the relationship between nasal conditioning of inhaled air and SARS-CoV-2 infectivity in the URT.

## 10. Conclusions

The importance of nasal air conditioning in SARS-CoV-2 infectivity remains poorly explored experimentally. The determination of SARS-CoV-2 infection rates and viral loads in the nasal cavity and nasopharynx in relation to the inhaled air temperature and humidity, age, gender, and genetic background is warranted. A better understanding of this may help to develop more effective measures to reduce infections and control the pandemic. Clinical studies on whether safe induction of type 1 and 3 IFNs in the URT reduces SARS-CoV-2 infection rates may be helpful to health personnel working in high risk situations. 

The importance of nasal air conditioning predicts that simple measures, like minimizing exposure to cold air and keeping the nose warm with a scarf wrapped around the face and neck or a face mask may help to promote more efficient nasal air conditioning to reduce infections. Keeping the nose warm in cold temperatures is an ancient and common practice to protect against respiratory illnesses in many parts of the world. While infectivity and immune protection in the URT is connected with the subsequent development of serious or even critical illness involving the lungs, many more factors, including comorbidities, come into play in the development of severe COVID-19. 

It is reasonable to conclude, however, that taking appropriate simple precautions to keep the nose warm to promote better nasal air conditioning in cold temperatures, particularly by more infection-susceptible persons, could minimise the initial infectivity of SARS-CoV-2 and other respiratory viruses. These measures promote more effective innate and adaptive immune responses in the URT. In the case of SARS-CoV-2, they supplement vaccination and previous infection with SARS-CoV-2 to further enhance adaptive immunity in the URT. The added advantage of using face masks and face scarves is that they also help reduce the person-to-person transmission of SARS-CoV-2 as well as other respiratory viruses.

## Figures and Tables

**Figure 1 ijms-22-07919-f001:**
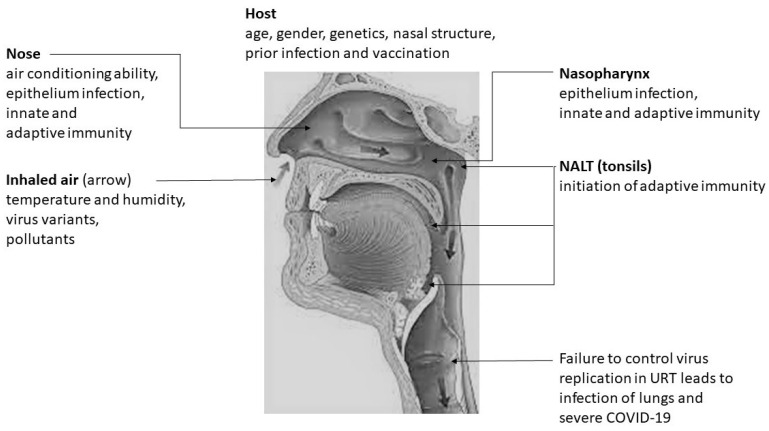
Relationship between the nasal conditioning of inhaled air and SARS-CoV-2 infectivity in the upper respiratory tract.

**Table 1 ijms-22-07919-t001:** Innate immune mechanisms that can protect against SARS-CoV-2 infection in the upper respiratory tract.

Induction	Effector Cell or Molecule	Effector Mechanism
-	Naturally occurring mucins, defensins and collectins	Bind virion and prevent cell binding and entry
Altered surface of the virion and virus-infected cells	Complement	Activation through the alternate or lectin pathway to promote lysis and opsonisation, inflammation
Pathogen associated molecular pattern (PAMP) recognition by pattern recognition receptors (PRRs)	Type 1 (α,β) and Type 3 (λ) interferons (IFNs)	Induction of anti-viral state in infected and neighbouring cells through inhibition of protein synthesis and mRNA degradation. Activation of phagocytic cells and dendritic cells
PRR	Inflammasome in macrophages and dendritic cells	Production of IL-1, IL-6 and TNF that promote an inflammatory response in tissue, fever and the synthesis of acute phase proteins
PRR	Macrophage and dendritic cell synthesis of IL-12, IL-18	Activation of NK cells to lyse virus infected cells and enhancement of adaptive immune response
Stress molecules expressed by infected cells	γδT cells secreting Type 2 IFNγ	Activation of NK cells, phagocytes, dendritic cells and the adaptive immune response

**Table 2 ijms-22-07919-t002:** Adaptive immune mechanisms that can protect against SARS-CoV-2 infection in the upper respiratory tract.

Effector Molecule or Cell	Mechanism of Action
Secreted IgA antibodies in mucus	Prevention of virion binding to epithelial cells by agglutination and neutralization of virions
IgG and IgM antibodies in mucosa and blood, including anti-A and anti-B blood group antibodies	Prevention of virion binding to host cells through agglutination and neutralization, activation of complement through the classical pathway, promoting opsonisation and phagocytosis, assisting NK cell killing through Fcγ receptors
CD4+ T_H_ lymphocytes	Activation of B cells, promoting immunoglobulin class switching and affinity maturation, secretion of cytokines like IFNγ that activate phagocytes and NK cells and upregulate major histocompatibility complex molecules.
CD8+ cytotoxic lymphocytes	Apoptosis of virus-infected cells by granzyme, perforin, etc.

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
