# Peer review of "Perspective of the Relationship between the Susceptibility to Initial SARS-CoV-2 Infectivity and Optimal Nasal Conditioning of Inhaled Air"

_ijms, 2021, doi:10.3390/ijms22157919_

Round 1

Reviewer 1 Report

The article is pretty well written. It's short, but full of meaningful content. 

I think it would be worth writing a few words about Sars-CoV2 infectivity prevention methods, based on the data presented in article.

Moreover, it is very interesting whether clinical trials are currently being conducted that confirm the data presented in this article.

Author Response

Reviewer 1

1.1 The article is pretty well written. It's short, but full of meaningful content.

Thank you

1.2 I think it would be worth writing a few words about Sars-CoV2 infectivity prevention methods, based on the data presented in article.

Section 10 [Conclusions] has been expanded in the revision to address this comment

1.3 Moreover, it is very interesting whether clinical trials are currently being conducted that confirm the data presented in this article.

I am not aware of any ongoing clinical trials and it is hoped that this article will stimulate such efforts that can prevent/minimise infectivity. One useful approach has been added in Section 10 [Conclusions]

Reviewer 2 Report

The paper aimed at highlighting the potential role of optimal nasal conditioning in SARS-COV2 infection. The subject matter is of interest however the manuscript should be carefully revised as it appears redundant and not focused.  In addition, several issues or are superficially described or missing.

Specific comments:

  • Several general issues reported in the background are widely known therefore it is not necessary to state them again
  • The paragraphs 2-3-4 should be merged
  • The section 5 should be extended taking into account the molecular and cellular findings on the mucosal response to SARS-CoV-2 in the nasal cavity and in Nasopharynx Associated Lymphoid Tissue System also analyzing its impact in the subsequent course of COVID-19.
  • In paragraph 8 the relationship between nasal air conditioning and genetic differences in susceptibility to SARS-CoV-2 infection should be better addressed
  • The impact of allergic diseases on nasal air conditioning and hence in COVID-19 susceptibility should be discussed.

Author Response

Reviewer 2

The paper aimed at highlighting the potential role of optimal nasal conditioning in SARS-COV2 infection. The subject matter is of interest however the manuscript should be carefully revised as it appears redundant and not focused.  In addition, several issues or are superficially described or missing.

The manuscript has been revised to address this.

Specific comments:

2.1 Several general issues reported in the background are widely known therefore it is not necessary to state them again.

In the International Journal for Molecular Sciences, the article will be read by many who are not virologists or familiar with SARS-CoV-2. Therefore, it is important to have this level of background to set the context and introduce specific terms and concepts used later in the article. The title of the background has been modified to reflect his.

2.2 The paragraphs 2-3-4 should be merged

Sections 2 and 3 have been merged as suggested and the merged section given a suitable title. The original section 4 (now section 3) deals with a distinct topic and has therefore retained.

2.3 The section 5 should be extended taking into account the molecular and cellular findings on the mucosal response to SARS-CoV-2 in the nasal cavity and in Nasopharynx Associated Lymphoid Tissue System also analyzing its impact in the subsequent course of COVID-19.

The Section (now Section 4) is clarified in the revision to address this comment. Relevant molecular and cellular findings in the immunology of SARS-CoV-2 infections in the URT are very limited at present and these are described in the MS. These together with data from influenza A enable us to make reasonable postulates about the mechanisms that are likely to operate in SARS-CoV-2 infections in the URT. Available immunological evidence reviewed in reference 10 is wholly compatible with what is stated in the MS, i.e. the poor control of SARS-CoV-2 replication in the URT leads to virus reaching the lungs through the oral-lung axis to cause severe COVID-19 where numerous other factors outside the aims of this article, including the involvement of different lymphoid structures, come into play.

2.4 In paragraph 8 the relationship between nasal air conditioning and genetic differences in susceptibility to SARS-CoV-2 infection should be better addressed

This is an important comment and the presentation has been improved in the revised MS. More detailed discussion is limited until additional experiments proposed in the article are performed.  

2.5 The impact of allergic diseases on nasal air conditioning and hence in COVID-19 susceptibility should be discussed.

This valuable suggestion is addressed by an added discussion on rhinitis in Section 9 of the revised MS with two additional references. Because the article is specifically restricted to susceptibility to early infection in the URT and not severe COVID-19 (although the former is a pre-requisite to the latter), asthma is only alluded to in the context of the effect of Type 2 cytokines on the URT. 

Reviewer 3 Report

This work critically reviewed a wide breadth of literature and provided relevant information regarding on the relationship between susceptibility to initial SARS-COV-2 infectivity and optimal nasal conditioning of in-haled air. And provided proper information on utilization of simple measures to control SARS-CoV-2 infectivity and illness. Specifically, the paper discussed

A well-written and informative paper, especially during this pandemic times. I would like to congratulate the author for providing the work. Wish to see more research work utilizing the suggestions provided in the paper. 

Minor comments:

Are these precautions along with vaccinations or in general to all viral infection or just SARS-CoV2? 

How about cold water showers activating the immune system? 

Including an accurate illustration showing the factors influencing the infectivity or the 5 points discussed in the paper will be beneficial. 

Author Response

Reviewer 3

This work critically reviewed a wide breadth of literature and provided relevant information regarding on the relationship between susceptibility to initial SARS-COV-2 infectivity and optimal nasal conditioning of in-haled air. And provided proper information on utilization of simple measures to control SARS-CoV-2 infectivity and illness. Specifically, the paper discussed

A well-written and informative paper, especially during this pandemic times. I would like to congratulate the author for providing the work. Wish to see more research work utilizing the suggestions provided in the paper.

Minor comments:

3.1 Are these precautions along with vaccinations or in general to all viral infection or just SARS-CoV2?

The precautions discussed are general for all respiratory viral infections. This has been clarified in the revised MS in Section 10. The same measures can continue to further reduce infectivity after vaccination. The relevant presentation has been improved in Section 10.

3.2 How about cold water showers activating the immune system?

Cold water showers traditionally believed to boost the immune system, but water temperature and exposure period are critical and can be deleterious if outside the narrow beneficial range. Valid experimental data on its relevance to nasal air conditioning and respiratory viral infections is lacking. A consideration of repetitive and variable exposure to cold water on SARS-CoV-2 infectivity is beyond the scope of this article.

3.3 Including an accurate illustration showing the factors influencing the infectivity or the 5 points discussed in the paper will be beneficial.

A new Figure 1 has been added to address this valuable suggestion.

Round 2

Reviewer 2 Report

The manuscript has been improved, therefore it is now suitable for publication